# Identification of Two Novel Linear B Cell Epitopes on the CD2v Protein of African Swine Fever Virus Using Monoclonal Antibodies

**DOI:** 10.3390/v15010131

**Published:** 2022-12-31

**Authors:** Wenting Jiang, Dawei Jiang, Lu Li, Jiabin Wang, Panpan Wang, Xuejian Shi, Qi Zhao, Boyuan Liu, Pengchao Ji, Gaiping Zhang

**Affiliations:** 1College of Veterinary Medicine, Henan Agricultural University, Zhengzhou 450046, China; jiang18860354181@163.com (W.J.); jiangdawei1010@126.com (D.J.); li18338215039@163.com (L.L.); wangjiabin0923@163.com (J.W.); wangpan11191001@163.com (P.W.); m18703610387@163.com (X.S.); m18838980392@163.com (Q.Z.); aka101lby@163.com (B.L.); 2International Joint Research Center of National Animal Immunology, Zhengzhou 450046, China; 3Longhu Laboratory, Zhengzhou 450046, China; 4Henan Engineering Laboratory of Animal Biological Products, Zhengzhou 450046, China

**Keywords:** African swine fever virus, CD2v, identification, B cell epitope, monoclonal antibody

## Abstract

African swine fever virus (ASFV) is a highly infectious viral pathogen that endangers the global pig industry, and no effective vaccine is available thus far. The CD2v protein is a glycoprotein on the outer envelope of ASFV, which mediates the transmission of the virus in the blood and recognition of the virus serotype, playing an important role in ASFV vaccine development and disease prevention. Here, we generated two specific monoclonal antibodies (mAbs), 6C11 and 8F12 (subtype IgG1/kappa-type), against the ASFV CD2v extracellular domain (CD2v-ex, GenBank: MK128995.1, 1–588 bp) and characterized their specificity. Peptide scanning technology was used to identify the epitopes recognized by mAbs 6C11 and 8F12. As a result, two novel B cell epitopes, ^38^DINGVSWN^45^ and ^134^GTNTNIY^140^, were defined. Amino acid sequence alignment showed that the defined epitopes were conserved in all referenced ASFV strains from various regions of China including the highly pathogenic, epidemic strain, Georgia2007/1 (NC_044959.2), with the same noted substitutions compared to the four foreign ASFV wild-type strains. This study provides important reference values for the design and development of an ASFV vaccine and useful biological materials for the functional study of the CD2v protein by deletion analysis.

## 1. Introduction

African swine fever (ASF) is caused by infection with African swine fever virus (ASFV), which is very infectious and often causes a fatal illness, seriously endangering the development of the pig industry [1,2]. ASFV is the only member of the Asfarviridae family [3] and the only known insect-borne DNA virus [4]. ASFV can cause persistent infection in soft ticks, which act as the natural host reservoir and transmission vector of ASFV [5,6,7]. ASFV can infect wild boar and domestic pigs with up to 100% lethality from highly pathogenic strains [8,9]. Since no effective vaccine or antiviral drugs are available at present, the prevention of ASFV mainly depends on detection and removal of all infected animals [10,11].

ASFV is a large double-stranded DNA virus with a genome of 170–193 kb, containing 151–167 open-reading frames (ORFs), encoding about 68 different structural proteins and more than 100 nonstructural proteins, the functions of about half of which are still unknown [3,5,12]. A total of 24 genotypes and 8 serotypes have been reported [13]. ASFV mainly invades monocyte/macrophages through clathrin-mediated endocytosis and actin-mediated micropinocytosis, and the invasion process is dependent on specific environmental conditions, including temperature, energetics, cholesterol and lower pH [14,15,16]. In addition, ASFV can achieve immune evasion by regulating the expression of cytokines and proinflammatory factors [17,18].

The outer envelope evolved from the host cytoplasm is formed during ASFV budding [9,19]. The CD2v protein, which is encoded by the *EP402R* gene, is found on the outer envelope of ASFV and has a molecular mass of 105–110 kDa after glycosylation [20]. CD2v is actively expressed in the late stage of viral infection, and its extracellular domain is a key mediator of the hemadsorption (HAD) process, which is involved in transport of the virus into the body [21,22,23]. Previous studies have shown that the CD2v protein can interact with surrounding lymphocytes and macrophages through lymphocyte function-associated antigen-3 (LFA-3/CD58) to promote NF-κB activation, inducing expression of interferon-β and interferon-stimulated genes, ultimately leading to apoptosis of lymphocytes and macrophages [24].

The naturally isolated attenuated ASFV strains show varying degrees of CD2v gene mutation or deletion [25,26,27]. Previous studies have shown that the live-attenuated ASFV vaccine constructed by deletion of the CD2v gene was safe and could induce cross-protection [28]. Further studies demonstrated that ASFV-SY18-∆CD2v/UK was nonpathogenic but was still able to elicit an antiviral immune response in pigs after vaccination [29]. Although live-attenuated vaccines pose a safety risk from virulence reversion, they are still considered as an ideal tool to unlock cross-protection [30]. The CD2v protein was also used in a subunit vaccine and live-viral vector vaccine and in subunit-DNA vaccine studies. Although the viral vaccines can induce specific antibodies and cellular immunity, how to provide optimal immune protection of pigs is still an unsolved problem [31,32,33].

In this study, BALB/c mice were immunized with the eukaryotic-expressed CD2v extracellular domain protein with glycosylation modification. Two specific monoclonal antibodies (mAbs) were obtained, and two conservative linear B cell epitopes (^38^DINGVSWN^45^ and ^134^GTNTNIY^140^) were targeted. This study provides the theoretical basis for design and development of a live-attenuated ASFV vaccine and subunit vaccine and supplies important biological materials for determining the mechanism of action of the CD2v protein and the differential identification of the ASFV wild-type strain and CD2v-deleted strain.

## 2. Materials and Methods

### 2.1. Proteins, Cells, and Plasmids

The recombinant CD2v extracellular domain (CD2v-ex, GenBank: MK128995.1, 1–588 bp) was prepared in our previous study [34], and the recombinant ASFV p30 protein with His-tag was maintained in our laboratory [35]. The SP2/0 myeloma cell line (ATCC, Manassas, VA, USA) was cultured in RPMI 1640 (Solarbio, Beijing, China) with 10% (*v*/*v*) fetal bovine serum (FBS; Gibco, Grand Island, NY, USA). The PK-15 cell line (ATCC, USA) was grown in Dulbecco’s modified Eagle’s medium (DMEM, Solarbio, Beijing, China) with 10% FBS. Cells were cultured at 37 °C in a humidified incubator (Thermo-Fisher Scientific, Waltham, MA, USA) containing 5% CO_2_. After codon optimization for the eukaryotic expression system, a 6× His-tag was added to the C-terminus. The full-length CD2v gene sequence (GenBank: MK128995.1) was directly cloned into the pcDNA3.1(+) vector digested with the restriction enzymes, NheI and EcoRI. The recombinant expression plasmid was synthesized by GenScript (Nanjing, China) and termed pcDNA3.1(+)-CD2v-His. The optimized CD2v-ex gene with a 6× His-tag at the C-terminus was cloned into the pcDNA3.1(+) vector and named pcDNA3.1(+)-CD2v-ex-His. The empty pcDNA3.1(+) vector was maintained by our laboratory.

### 2.2. Production of mAbs against ASFV CD2v Protein

Female 6–8 weeks old BALB/c mice were obtained from the animal experimental center of Huaxing (Zhengzhou, China) and housed in specific pathogen-free (SPF) isolators under negative pressure. Food and water were provided ad libitum. Animals were injected subcutaneously with 15 μg of recombinant CD2v-ex emulsified with an equal volume of complete Freund’ s adjuvant (Sigma, St.Louis, MO, USA). Mice were boosted twice by immunization with protein in incomplete Freund’s adjuvant (Sigma, St. Louis, MO, USA). The interval between each immunization was two weeks. The antibody levels were measured by indirect ELISA (iELISA) using recombinant CD2v-ex. An intraperitoneal booster with twice the amount of protein (30 μg/mouse) without Freund’s was given three days prior to cell fusion for the mouse with the highest antibody level.

To select hybridoma cells secreting mAbs against the CD2v protein, single splenocytes were aseptically isolated from selected euthanized mice and fused with SP2/0 myeloma cells at a ratio of about 2:1 using PEG1500 (Sigma) following a standard procedure [36]. The fused cells were seeded in 96-well plates in culture medium containing hypoxanthine–aminopterin–thymidine (HAT) (Sigma) for hybridoma selection at 37 °C in 5% CO_2_. On days 7 and 9, the medium was replaced with fresh HAT-medium. The mAbs against CD2v secreted by the hybridoma cells were identified by CD2v-ex-iELISA using a method adapted from a previous study [37]. The preliminarily identified positive cells were subcloned after screening culture in medium containing hypoxanthine–thymidine (HT) (Sigma). To isolate a purified hybridoma cell line, the positive hybridomas were subcloned three times by limiting dilution. CD2v-ex-iELISA and p30-iELISA were performed to select mAbs that specifically recognized the CD2v-ex protein rather than the His-tag. The Ig class of the hybridomas was determined with a mouse mAb isotype ELISA kit (Proteintech, Wuhan, China).

### 2.3. Western Blotting for CD2v-ex Protein

Western blotting analysis was run to determine the immunoreactivity of the mAbs with natural recombinant CD2v-ex protein. The purified CD2-ex protein was mixed with 5× loading buffer (Solarbio), denatured by heating at 98 °C for 10 min, separated by SDS-PAGE on a 7.5% gel (Epizyme, Shanghai, China), and then blotted to a 0.45 µm polyvinylidene fluoride (PVDF) membrane (Millipore, Billerica, MA, USA). After blocking with 5% (*w*/*v*) skim milk (SM) in PBST containing 0.05% (*v*/*v*) Tween-20 for 1 h at 37 °C, the PVDF membranes were incubated with hybridoma culture supernatants for 1 h at room temperature (RT) with supernatants from SP2/0 cultures as negative control and positive anti-ASFV sera (1:2000 in 5% SM, *v*/*v*; China Veterinary Culture Collection Center, CVCC, China) as positive control. After washing five times (5 min each) with PBST, membranes were incubated with HRP-conjugated goat anti-mouse IgG (H + L) (1:5000 in 5% SM, *v*/*v*; Proteintech, Wuhan, China) for 1 h at RT, while HRP-conjugated mouse anti-swine IgG (H + L) (1:5000 in 5% SM, *v*/*v*; Immunoway, Plano, TX, USA) was used against ASFV-positive serum. After the final five washes, the enhanced chemiluminescence (ECL) Western blotting substrate BeyoECL Plus (Beyotime, Shanghai, China) was added and signals were visualized with a multi-functional imaging system (GE-Amersham Imager600, Boston, MA, USA).

### 2.4. Hemadsorption (HAD) and Indirect Immunofluorescence Analysis (IFA)

To identify the location of the CD2v protein in CD2v-transfected PK-15 cells, the cells were grown in 24-well plates and transfected with recombinant plasmids pcDNA3.1(+)-CD2v-His and pcDNA3.1(+)-CD2v-ex-His at a concentration of 0.5 μg/well using Lipofectamine-2000 (Invitrogen, Carlsbad, CA, USA); cells transfected with the empty vector, pcDNA3.1(+), were used as negative control. At 30 h post-transfection, fresh suspensions of healthy pig erythrocytes (1%, *v*/*v*, in 1× PBS, 100 µL/well) were inoculated into the plate, and incubation was continued for 6 h. After washing off the non-adsorbed erythrocytes with sterile PBS, the HAD phenomenon was observed. The transfected cells were fixed with 4% (*m*/*v*) paraformaldehyde (PFA) solution for 30 min at RT and then blocked with 5% (*w*/*v*) bovine serum albumin (BSA) in PBS for 1 h at 37 °C, the cells were incubated with anti-CD2v-ex mouse polyclonal antibodies (1:500 in 5% BSA, isolated from the blood of mice immunized with CD2v-ex protein before cell fusion) for the HAD assay 1 h at 37 °C. To determine the specificity of selected mAbs against CD2v, IFA tests were performed by adding culture supernatants of hybridoma cells to fixed CD2v-transfected PK-15 cells. Blank SP2/0 cell culture supernatants and anti-CD2v-ex mouse polyclonal antibodies were used as the negative and positive controls, respectively. FITC-conjugated goat anti-mouse IgG (H + L, 1:500, in 5% BSA, Proteintech) was used as secondary antibody. DNA was stained with 4′,6-diamidino-2-phenylindole (DAPI; Beyotime). Fluorescence signals were detected using an LSM 800 laser-scanning confocal microscope (Zeiss, Germany).

### 2.5. Blocking/Competitive ELISA Analysis of ASFV Positive Serum

To evaluate the production of the CD2v epitopes targeted by mAbs in positive anti-ASFV sera (CVCC, China), a blocking/competitive ELISA was performed. Purified recombinant CD2v-ex protein was added to carbonate–bicarbonate buffer (CBS buffer, pH 9.6) at a concentration of 4 μg/mL (400 ng/well) and used to coat 96-well plates (Beaver, Suzhou, China) for 12 h at 4 °C. Coated plates were washed five times with PBST and shaken dry. After blocking for 1 h at RT with 5% SM and washing as above, mixed ASFV-positive serum (1:2 in 1% BSA in PBS, 50 μL/well), culture supernatants of hybridoma cells, or 50 μL/well, SP2/0 cell culture supernatant as control, were added to plates and incubated 1 h at 37 °C. Negative anti-ASFV sera (kept in our laboratory and determined by Kernal-p30-iELISA kit) with the same treatment was used as negative control. The plates were washed and incubated with HRP-conjugated goat anti-mouse IgG (Proteintech, Wuhan, China) diluted 1:5000 with 5% SM. After 1 h at 37 °C, the plates were washed, and 3,3′,5,5′-tetramethylbenzidine (TMB, Solarbio) was added for 10 min at RT. The reaction was stopped by addition of 3 mol/L HCl (50 μL/well). The optical density (OD) was measured at 450 nm (OD_450_) with a multimode microplate reader (Tecan 10 M, Zürich, Switzerland). Each reaction had three replicates. The absorbance values were converted to percent inhibition (PI) by: PI (%) = [1 − (test sample OD_450_/negative sample OD_450_] × 100% [38].

### 2.6. Identification of Linear B Cell Epitopes

To detect the epitopes on CD2v that are recognized by the identified mAbs, according to bioinformatics website prediction (http://tools.iedb.org/bcell/result/, accessed on 19 February 2022), peptide-scanning was performed first to select 19 overlapping synthetic peptides corresponding to the truncated CD2v-ex protein (14–196 amino acids, the signal peptide sequence was not included). Each synthetic peptide overlapped the previous peptide by 4 amino acids. All peptides were synthesized and conjugated with IgG-free BSA by Genescript. The peptides were used as detection antigens in dot blotting and iELISA assay to identify the epitopes recognized by the identified mAbs [39]. For dot blotting, peptides were diluted with ultrapure water to the recommended concentration of 0.1 mg/mL and blotted onto nitrocellulose (NC) membranes (2.5 μL/dot); PBS and purified CD2v-ex protein (0.1 mg/mL) were blotted as negative and positive controls, respectively. After drying naturally and blocking with 5% SM, NC membranes were incubated with supernatants from hybridoma cultures or SP2/0 cells (negative control). After washing with PBST, the secondary antibody incubation and reaction visualization were performed as described for Western blotting. Briefly, for iELISA, peptides were diluted with PBS to a concentration of 10 μg/mL, and plates were coated overnight at 4 °C. PBS and purified CD2v-ex protein (200 ng/well) were used as negative and positive controls, respectively. After blocking with 5% SM, plates were incubated with hybridoma supernatants or SP2/0 cells (negative control) for 1 h at 37 °C and then with HRP-conjugated goat anti-mouse IgG (Proteintech, 1:5000 in 5% SM) for 1 h at 37 °C. After addition of TMB solution for 10 min at RT, reactions were stopped with 3 mol/L HCl (50 μL/well), and OD_450_ was read with a microplate reader (Tecan 10 M). Each reaction consisted of three replicates.

After initial identification of the CD2v peptides recognized by mAbs, they were cleaved into fragments according to the predicted results to determine the positions of the functional B cell epitopes. Cysteines were added to all shorter peptides for better coupling at the C-terminus, synthesized by Shanghai Apeptide Co., Ltd., and then crosslinked with BSA using 4-(N-maleimidomethyl) cyclohexane-1-carboxylic acid 3-sulfo-N-hydroxysuccinimide ester (SMCC, Yuanye, Shanghai, China). Shorter peptides conjugated to IgG-free bovine serum albumin were used as detection antigens in dot blotting and iELISA assays to verify recognition by mAbs as described above. PBS with the same treatment and purified CD2v-ex protein (200 ng/well) were set as negative and positive controls, respectively.

### 2.7. B Cell Epitope Conservation

To measure the conservation of identified epitopes, 13 reference ER402R gene sequences of wild ASFV strains from NCBI were downloaded for alignment, including eight strains from various regions in China (MN172368.1, MN393476.1, MK333180.1, MK333181.1, MK645909.1, MN393477.1, MT496893.1, MK940252.1) and five foreign epidemic strains (NC_044947.1, NC_044959.2, NC_044958.1, NC_001659.2, NC_044943.1). The transmembrane sequences were first predicted using the online tool TMHMM-2.0 (https://services.healthtech.dtu.dk/service.php?TMHMM-2.0, accessed on 20 November 2021), and then, the extracellular domain amino acid sequences were aligned with the target CD2v-ex sequences using DNAman 6.0 software (Lynnon BioSoft Inc., CA, USA) to determine conservation of the defined B cell epitopes.

### 2.8. Statistical Analysis

All statistical analyses were conducted using GraphPad Prism 8.2 software (Graph Pad Prism Inc., San Diego, CA, USA).

## 3. Results

### 3.1. Production and Characterization of CD2v Protein-Specific Monoclonal Antibodies

To generate anti-CD2v mAbs, three BALB/c mice (6–8 weeks) were inoculated with recombinant CD2v-ex protein. After fusion, the hybridomas were tested by CD2v-ex-iELISA. After subcloning three times, two hybridoma cell lines named 6C11 and 8F12 were obtained. The results of Western blotting showed a specific band (60–90 kDa) (Figure 1A), indicating that mAbs 6C11 and 8F12 specifically recognized the linear epitope of the recombinant CD2v-ex protein. The two mAbs showed positive results to CD2v and negative results to p30 in iELISA (Figure 1B), indicating that the selected mAbs specifically recognized CD2v-ex protein and not the His-tag. Isotype identification revealed that mAbs 6C11 and 8F12 were an IgG1/kappa subtype (Figure 1C).

### 3.2. Localization of CD2v Protein in Transfected Cells

To determine where CD2v protein was localized, CD2v-tranfected PK-15 cells were fixed and incubated with anti-CD2v-ex mouse polyclonal antibody or hybridoma supernatants and then with FITC-conjugated goat anti-mouse IgG (H + L) and DAPI. As shown in Figure 2A, the HAD phenomenon was observed in CD2v-transfected PK-15 cells, but not in CD2v-ex-transfected cells. The IFA experiments showed that the erythrocytes bound to the surface of CD2v-transfected PK-15 cells were stained green, while the staining sites of CD2v-ex-transfected cells were concentrated in the cytoplasm, indicating that the full-length expression of the CD2v protein can cause HAD and colocalization in cytomembranes. This phenomenon confirmed previous studies showing that the intracellular domain of CD2v might be related to protein localization [20,40]. In addition, regarding IFA tests using mAbs 6C11 and 8F12 as the first antibodies, the results showed green fluorescence in the cytoplasm, consistent with the positive control (Figure 2B), indicating that the two mAbs could specifically recognize the CD2v protein that colocalized in the cytoplasm.

### 3.3. Reactivity of ASFV-Positive Serum against Anti-CD2v Monoclonal Antibodies

To determine the blocking effect of mAbs 6C11 and 8F12 against positive anti-ASFV sera, a blocking ELISA using a plate coated with recombinant CD2v-ex protein was performed using the selected mAbs as blocking antibody. The results showed that the PIs of mAbs against positive anti-ASFV sera were both >50% (Figure 3), proving that the mAbs could block the binding of positive anti-ASFV sera to the CD2v protein. This suggested that the epitopes recognized by 6C11 and 8F12 could generate strong B cell immunity in ASFV-infected swine.

### 3.4. Precise Localization of the mAb Epitopes

To pinpoint the epitopes recognized by the mAbs, the CD2v-ex amino acid sequence was first truncated to 19 overlapping peptides according to the prediction results of the bioinformatics website, and 15 soluble peptides were successfully synthesized (Figure 4A). The results of dot blotting and iELISA showed that mAb 6C11 could identify peptide No.3: ^34^NDNNDINGVSWNF^46^ and mAb 8F12 could identify peptide No.13: ^130^KKNNGTNTNIYLNIN^144^ (Figure 4B,C). To identify the precise linear epitopes, shorter overlapping peptides spanning the identified peptides No.3 and No.13 were further synthesized according to the bioinformatics website prediction results. For peptide No.3, the shorter peptide sequences are listed in Figure 4D. Dot blotting and iELISA results showed that the mAb 6C11 could identify the peptides 3.4: ^37^NDINGVSWN^45^ and 3.5: ^38^DINGVSWNF^46^ but could not identify the peptides 3.5.1: ^38^DINGVSW^44^ and 3.5.2: ^39^INGVSWN^45^ (Figure 4E–H), indicating that mAb 6C11 could identify a liner B cell epitope of ^38^DINGVSWN^45^. For peptide No.13, the shorter peptide sequences are listed in Figure 4I. Dot blotting and iELISA results showed that the short peptide 13-5: ^134^GTNTNIY^140^ was the linear B cell epitope of the CD2v protein (Figure 4J–M).

### 3.5. Sequence Alignment Analysis of Epitope Conservation

Sequence alignment was performed to explore the level of conservation of the epitope of mAbs 6C11and 8F12 among 13 domestic and foreign ASFV wild strains selected from NCBI using DNAman software. As the alignment results show in Figure 5, the epitopes identified by mAbs 6C11(^38^DINGVSWN^45^) and 8F12 (^134^GTNTNIY^140^) were highly conserved among the eight domestic and one ASFV strain, with shared sequence similarity of 100%, but compared with the four foreign strains, the same substitutions were noted.

## 4. Discussion

ASFV was first reported in China in 2018 and identified as p72 genotype II and CD2v serotype 8, a homolog of the pathogenic epidemic strain Georgia 2007 [40], causing huge economic losses in the pig industry. Due to its characteristics of strong pathogenicity and high mortality, removing infected animals is still the most effective control method [41]. However, recent studies found that the natural variant epidemic strain of ASF gene type II showed a non-HAD phenotype, which was manifested by the loss of the CD2v protein, some of the variant epidemic strains have low mortality, and the emergence of natural weak virus strains has caused new problems for the early diagnosis, epidemic control, and virus purification of ASF [27].

The CD2v protein is expressed on cells and is located around the perinuclear virus factory. When there is a corresponding receptor outside the cell, the CD2v protein is transported from the cell to the cell membrane surface to bind to the receptor [20], and this phenomenon has been verified in our experiments. However, the mechanism underlying this activity has not been clarified. Because the expression of the CD2v protein in various virus strains is different, its impact on the virulence of the virus needs further study [28,42,43]. At present, the serodiagnosis of ASFV is mainly based on antibody detection methods of p72, p30 and p54 proteins [44,45,46], and there is no commercial kit with the CD2v protein as the diagnostic target. The development of efficient vaccines can reduce ASFV transmission [47].

Developing a safe and effective ASFV vaccine is still a worldwide problem. The classical heat-inactivated whole virus has not proved feasible, and research on subunit, live vector-based, DNA, and live-attenuated vaccines (LAVs) is in progress [48,49]. LAVs are considered the most promising type of vaccine to be developed successfully among these tested vaccines [26,50,51]. LAVs can be developed by screening naturally attenuated virus isolates and by cell passage and deletion of the virulence-associated gene(s) [52]. However, abnormal expression of the CD2v protein has been observed in some naturally isolated attenuated strains [25,53,54]. In addition, individual or combined deletions of CD2v can attenuate virulent ASFV isolates and induce protective immune responses against virulent parental virus challenge in pigs [28,42,55,56].These results show the importance of CD2v in the design and development of LAVs and suggest that ASFV LAVs with gene deletions may be the most promising ASFV vaccines [57]. However, for the selection and combination of target gene(s) and original strains, a large number of clinical trials are needed.

In this paper, we produced two high-affinity specific mAbs against the CD2v-ex protein and found two novel linear B cell epitopes, ^38^DINGVSWN^45^ and ^134^GTNTNIY^140^. The mAbs 6C11 and 8F12 could both specifically recognize the full-length recombinant CD2v protein, and the combination of CD2v-ex and mAbs (6C11 and 8F12) could be blocked by ASFV-infected swine serum. The defined two epitopes were conserved in all referenced ASFV strains from various regions of China and the highly pathogenic epidemic strain Georgia2007/1 (NC_044959.2), but with the same noted substitutions compared with the four foreign ASFV wild strains (NC_044947.1, NC_044958.1, NC_001659.2, NC_044943.1). We believe that the epitopes have important reference value for the design and development of ASFV vaccines and that mAbs 6C11 and 8F12 can be used to verify infection with the ASFV wild type or CD2v-deletion strain by blocking ELISA. They can also be used to explore the mechanism of CD2v protein action.

## Figures and Tables

**Figure 1 viruses-15-00131-f001:**
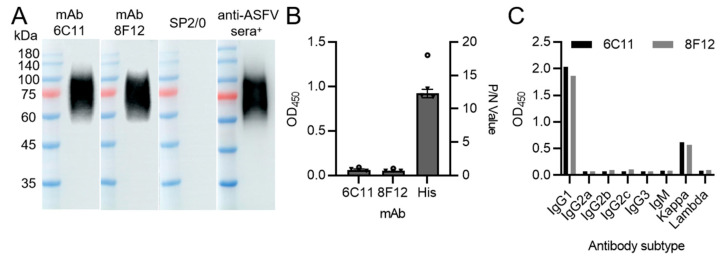
Production and characterization of two CD2v protein-specific mAbs. (**A**) Western blotting results of mAbs 6C11 and 8F12 reaction against CD2v-ex protein. SP2/0 cell supernatants and positive anti-ASFV sera (anti-ASFV sera^+^) were used as negative and positive controls, respectively. (**B**) Indirect ELISA results of mAbs 6C11 and 8F12 reaction against p30 protein with His-tag. His mAb was used as positive control. Data are presented as mean ± SD. (**C**) Isotype determination of mAbs 6C11 and 8F12.

**Figure 2 viruses-15-00131-f002:**
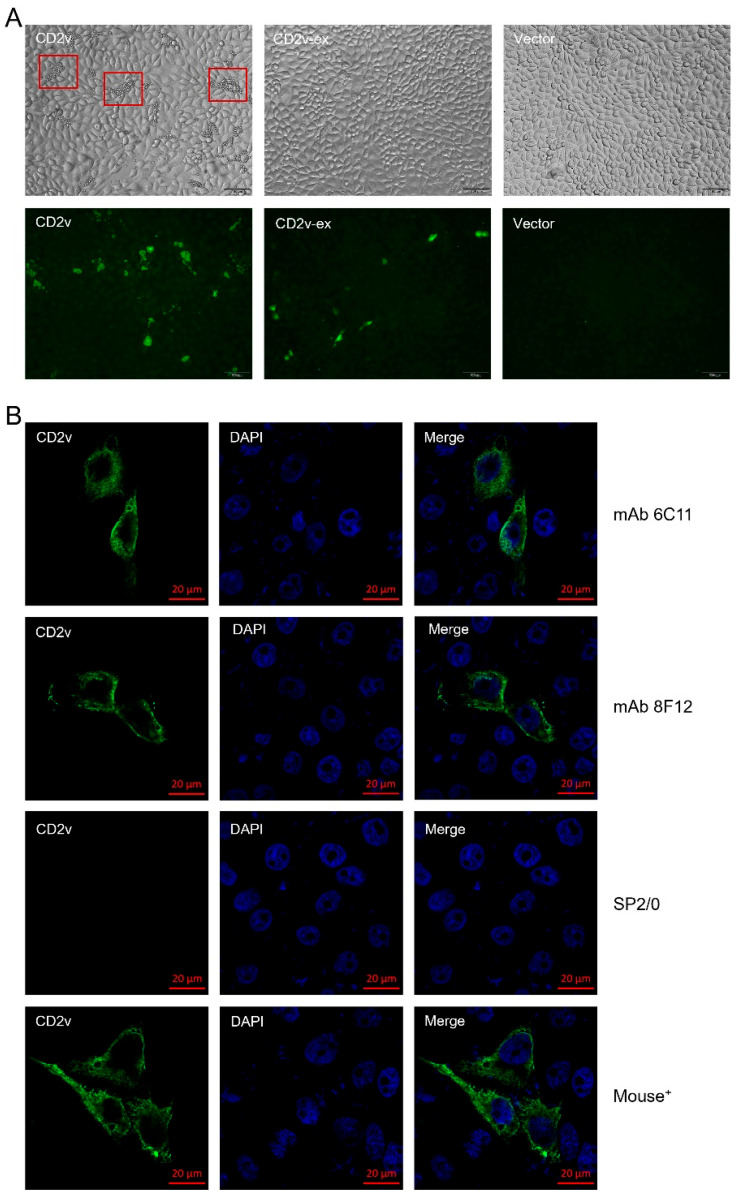
Localization of CD2v protein in transfected cells. (**A**) HAD test results of CD2v protein in transfected PK-15 cells. (**B**) IFA test results of mAbs 6C11 and 8F12 against CD2v. SP2/0 cell supernatants and anti-CD2v-ex mouse polyclonal antibodies (Mouse^+^) were used as negative and positive controls, respectively.

**Figure 3 viruses-15-00131-f003:**
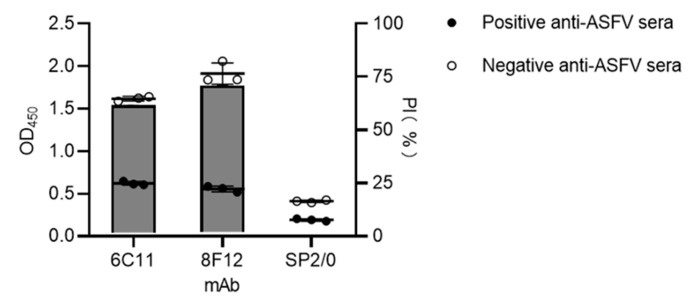
Blocking/competitive ELISA detection of positive anti-ASFV sera against mAbs 6C11 and 8F12. Negative anti-ASFV sera was set to determine negative sample OD_450_ value of blocking/competitive ELISA. SP2/0 cell supernatants were used as negative control. Data are presented as mean ± SD.

**Figure 4 viruses-15-00131-f004:**
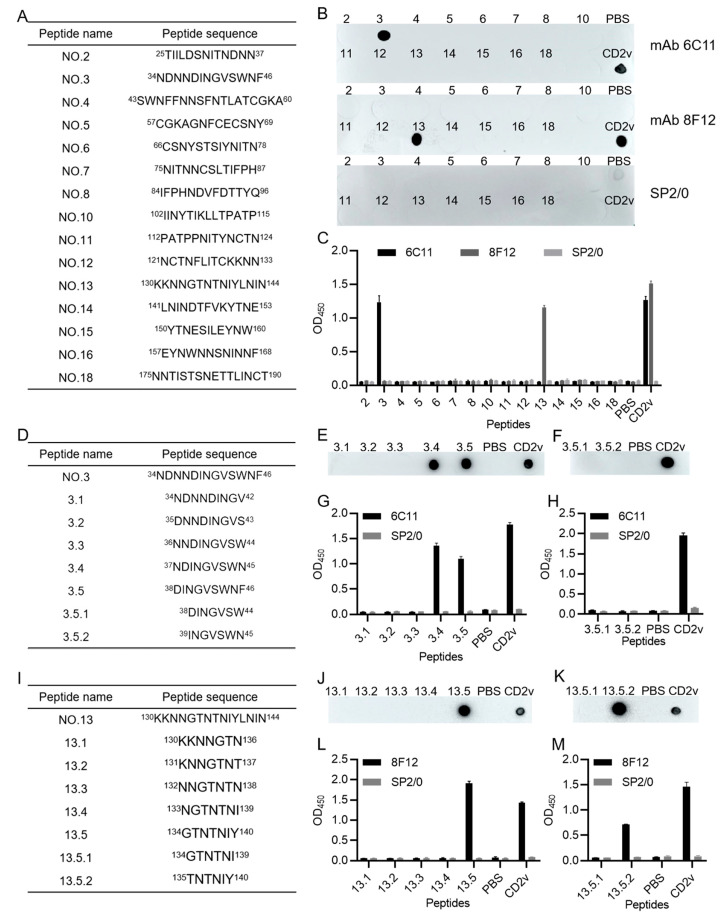
Identification of the epitopes recognized by mAbs 6C11 and 8F12. (**A**) Fifteen synthetic overlap ping peptides spanning the CD2v protein. (**B**,**C**) Dot blotting and indirect ELISA analysis of mAbs 6C11 and 8F12 against the fifteen peptides. SP2/0 cell supernatants were used as negative control. (**D**) Seven shorter overlapping peptides spanning peptide No.3: ^34^NDNNDINGVSWNF^46^. (**E**–**H**) Dot blotting and indirect ELISA analysis of mAb 6C11 against the seven shorter peptides from peptide No.3. (**I**) Seven shorter overlapping peptides spanning peptide No.13: ^130^KKNNGTNTNIYLNIN^144^. (**J**–**M**) Dot blotting and indirect ELISA analysis of mAb 8F12 against the seven shorter peptides from peptide No.13. PBS and purified CD2v protein were set used as negative and positive controls, respectively.

**Figure 5 viruses-15-00131-f005:**
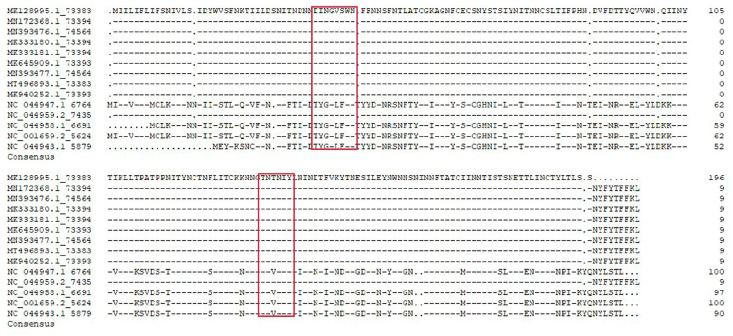
Conservative analysis of the defined epitope sequences.

## Data Availability

All available data are presented in this manuscript.

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
