# Peer review of "Identification of Two Novel Linear B Cell Epitopes on the CD2v Protein of African Swine Fever Virus Using Monoclonal Antibodies"

_viruses, 2022, doi:10.3390/v15010131_

Round 1
Reviewer 1 Report
Comments on viruses-2106731
Identification of two novel linear B cell epitopes on the CD2v protein of African swine fever virus using monoclonal antibodies
The present study by Wenting Jiang et al. reported two novel conserved linear B-cell epitopes on the CD2v protein of African swine fever virus (ASFV). The work shows something new. However, the study design and presentation should be improved. Several major concerns should be addressed.
1. The key materials, including the anti-CD2v/ASFV sera, should be described in more details.
2. Hemadsorption-inhibition assay should be performed to show the inhibitory effects of the two anti-CD2v monoclonal antibodies on ASFV hemadsorption in porcine alveolar macrophages .
3. The reactivity of the identified B-cell epitopes with a panel of anti-ASFV porcine sera against different genotypes of ASFV should be examined.
4. The presentation of the tables and figures should be improved.
5. The manuscript should be revised by native English speakers.
Reviewer 2 Report
In general the article prepared by author is interesting. The high number of analysis is a big advantage for the readers. Article has merit however can be improved. The results section is written very well but the discussion is not sufficient. In that part authors did not present enough the current data connected with CD2v. There should be some reference to LAVs and the probability of the use of their results in future vaccine. Paper has merit but the the discussion must be enhance. In present form did not respond to the purpose of the study.
From minor things I have notice:
Line 5 – there is no affiliation with one of the authors.
Line 38 – lack of reference.
Sincerely,
Reviewer
Round 2
Reviewer 1 Report
The revised manuscript has been improved moderately. The manuscript writing needs to be modified further.
ASFV+ sera=ASFV-containing sera, which should be differentiated from anti-ASFV sera.
ASFV-negative (ASFV–) or ASFV-positive (ASFV+) sera are not correct and should be changed as negative or positive sera against ASFV (or antisera against ASFV or anti-ASFV sera).
Author Response
Dear Reviewer:
Thank you again for your further comments concerning our manuscript entitled “Identification of two novel linear B cell epitopes on the CD2v protein of African swine fever virus using monoclonal antibodies” (Manuscript ID: viruses-2106731). These comments were valuable and helpful for revising and improving our manuscript. We have studied the comments carefully and have made revisions and corrections (marked using the“Track Changes” in the revised manuscript), which we hope will meet with your approval.
To facilitate this discussion, we retyped your comments in blue and then presented our responses to the comments.
Comment 1:
ASFV+ sera=ASFV-containing sera, which should be differentiated from anti-ASFV sera.
Response 1:
Thank you for pointing out this issue. We have benefited a lot and will use the two descriptions differently in further writing.
Comment 2:
ASFV-negative (ASFV–) or ASFV-positive (ASFV+) sera are not correct and should be changed as negative or positive sera against ASFV (or antisera against ASFV or anti-ASFV sera).
Response 2:
Thank you for pointing out this problem; we apologize for our incorrect statement. We have changed all the description of “ASFV-negative (ASFV–) or ASFV-positive (ASFV+) sera” with “negative or positive anti-ASFV sera” (line 139, 172, 180, 255, 282, 285, 286, 290, 291 and Figure 1, 2).
Special thanks to you for your valuable comments.
Best regards,
Pengchao Ji
